# Application of Honey to Reduce Perineal Laceration Pain during the Postpartum Period: A Randomized Controlled Trial

**DOI:** 10.3390/healthcare10081515

**Published:** 2022-08-11

**Authors:** Désirée Gerosa, Marika Santagata, Begoña Martinez de Tejada, Marie-Julia Guittier

**Affiliations:** 1School of Health Sciences Geneva, HES-SO University of Applied Sciences and Arts Western Switzerland, 1206 Geneva, Switzerland; 2Obstetrics’ Division, Department of Pediatrics, Gynecology and Obstetrics, University Hospitals of Geneva, 1206 Geneva, Switzerland; 3Faculty of Medicine, University of Geneva, 1205 Geneva, Switzerland

**Keywords:** perineum, tears, episiotomy, honey, pain, postpartum, childbirth, randomized controlled trial

## Abstract

Perineal lacerations affect between 35 and 85% of women during childbirth and may be responsible for postpartum pain. Honey has been demonstrated to have interesting properties that can promote wound healing. The aim was to evaluate the effectiveness of the application of honey to the perineum to reduce perineal pain during the early postpartum period. A randomized controlled trial including 68 women was conducted. In the intervention group, honey was applied to perineal lacerations for four days, in addition to standard care. The control group received only standard care. The primary outcome was pain intensity using the Visual Analog Scale and pain perception using the McGill Pain Questionnaire (QDSA). The secondary outcomes were a burning sensation, the use of a pain killer, and the women’s satisfaction with the honey application. The intensity of pain was not significantly different between the groups on Day 1 (VAS 3.38 in the control group versus 3.34 in the intervention group, *p* = 0.65) or on Day 4 (VAS 2.28 versus 1.41, respectively, *p* = 0.09). There was no significant difference regarding the perception of pain with the QDSA. Despite this, most of the women in the intervention group (93%) were satisfied or very satisfied with the use of honey on their perineum.

## 1. Introduction

The prevalence of perineal laceration following vaginal birth ranges from 35% to 85% in European countries, with primiparous women being affected more frequently than multiparous women [1,2]. Perineal lacerations include four degrees of perineal tears, episiotomies, and anterior vulvar tears. Some obstetrical variables have been shown to be predictive of anal sphincter injuries (vacuum extraction and fetal weight exceeding 4000 g) and/or second-degree tear (vacuum extraction). Vacuum extraction, fetal head circumference exceeding 35 cm, and heredity of pelvic floor dysfunction and/or connective tissue deficiency were associated with an increased risk of high vaginal tears [1]. Such lesions may be responsible for short-, medium-, and long-term inconveniences. Regarding the physical aspects, women have reported dyspareunia (54%), flatulence incontinence (28%), fecal incontinence (20%), stress urinary incontinence (17%), and urge incontinence (11%) [3,4]. According to the same authors, 4% of women report symptoms of prolapse, described as sensations such as tissue protrusions, vaginal pressure, or pain. Infection or dehiscence of the scar occurs in about 6% of cases [5]. Six weeks postnatal, 20% of women still have pain following perineal trauma [6]. This morbidity secondary to vaginal delivery has a negative impact on self-esteem and quality of life after childbirth [7,8]. The maternal psychological well-being is also affected, causing fatigue, stress, anxiety, and fear of future birth [7,8]. These issues may negatively affect early maternal bonding and breastfeeding. In addition, pain relief drugs are used widely during the postpartum period to treat the pain of various origins, such as uterine trenches, breastfeeding, and perineal trauma [9]. These drugs can be responsible for maternal side effects and can be found in maternal milk [10]. Reducing pain and, therefore, the use of such medication could be beneficial for mothers and newborns. Promoting the protection of the perineum during childbirth and perineal care during the postpartum period is therefore important.

Honey has been shown to have antibacterial and anti-inflammatory properties [11]. Honey is composed of above 80% sugar, 18% water, and small percentages of other molecules. Its low percentage of water, the presence of hydrogen peroxide, and pH between 3.2 and 5.5 interfere with bacterial growth. Furthermore, its composition creates an osmotic flow that drains the liquid around the wound and reduces inflammation [12]. There are a small number of studies evaluating the effects of honey on perineal trauma. Only one completed study from Iran has investigated the effect of honey on episiotomy-related pain after vaginal birth [13]. The authors assessed the level of pain with a Likert scale. The results showed no significant decrease in pain intensity on Day 7 and Day 14 postpartum. A randomized controlled trial (RCT) in India showed a benefit in the perineal wound healing process from the application of honey in comparison with Betadine^®^ [14]. The authors used a scale evaluating five dimensions (REEDA for redness, edema, ecchymosis, discharge, and approximation of the perineal tissues). The lower the score is, the better the wound healing is. The mean value on the scale on Day 2 was 4.53 (SD 2.16) for the honey group and 6.63 (SD 1.94) for the Betadine^®^ group, *p* = 0.00. At Day 5, the results were 0.27 (SD 0.64) vs. 1.40 (SD 1.33), respectively, *p* = 0.00.

We designed an RCT whose aim was to evaluate the effectiveness of the application of medical honey on perineal laceration after childbirth in reducing perineal pain during the early postpartum period.

## 2. Materials and Methods

### 2.1. Trial Design and Setting

A randomized controlled trial including 68 women was conducted from September 2018 to January 2019. The participants were split into two groups: honey application on the perineum and standard care (intervention group) versus only standard care (control group).

This study was conducted in the maternity hospital of the University Hospital of Geneva. This is the biggest maternity ward in Switzerland, with more than 4000 childbirths occurring here per year. The rates of perineal lacerations in this maternity hospital are similar to those reported in the literature, with approximately 61% of births resulting in perineal injuries, with 22% of women experiencing first-degree tears, 28% experiencing second-degree tears, 2% experiencing third-degree tears, 0.1% experiencing fourth-degree tears, and 13% experiencing mid-lateral episiotomies.

### 2.2. Sample Size

Based on the existing literature, which uses the VAS (0–10) to assess perineal pain [15,16], we considered a difference of 1.5 points to be clinically relevant. For the sample size calculation, we expected mean levels of pain to be 2.5 for the experimental group and 4 for the control group. In the reference studies for the calculation, standard deviations (SD) of the VAS distribution ranged from 1.5 to 2.2 [17,18]. In view of these results, we considered a medium SD of 2.0. A significance alpha threshold of 0.05 and a statistical power of 80% was chosen for a two-tailed test. With these settings, this RCT required 56 women, with 28 in each group [19]. To compensate for any loss to follow-up, we increased the number of participants by 10% to account for a drop-out rate of 10% for a total of 62 women. We planned on recruiting 62 women (31 per group).

Randomization was carried out after the verification of the inclusion and exclusion criteria and after signing the consent form. It was generated using a computer program with blocks of 4, 6, and 8. The randomization was stratified based on the presence or absence of grazes (anterior vulvar tears), assuming that an imbalance could cause bias between groups. Indeed, as these wounds are generally not sutured, their healing time may be longer [20]. The ratio for “honey application” versus “no honey application” was 1:1. Randomization was carried out using sealed and opaque envelopes prepared in advance with a numbering system. The preparation of these envelopes was carried out by a secretary of the obstetric research unit with the blinding of the authors. Each envelope included the affiliated group and a reference code to ensure anonymity.

### 2.3. Participants and Recruitment Procedure

The inclusion criteria for women were: over 18 years of age; undergoing vaginal birth; good ability to understand and read French; and the presence of episiotomies or first- and second-degree perineal tears, whether associated with grazes or not.

A first-degree perineal tear was defined as a vaginal and/or cutaneous lesion, and a second-degree tear as one affecting the vaginal mucosa, the skin, and the superficial muscles of the perineum [21]. Exclusion criteria were: third- and fourth-degree tears because of their low prevalence and specific management requirements, postpartum hemorrhage defined as >500 milliliters of blood loss because it can interfere with the healing process [22], and allergy to honey or bee venom.

Recruitment was performed by two midwives on the day of childbirth (defined as Day 0, D0) in the postpartum unit between 2 and 24 h postpartum. Recruitment during the postpartum period and not in the childbirth room allowed the women to recover emotionally from the first moments with their newborns and to be more cognizant during the explanation of the study. After checking whether they met the inclusion criteria or any of the exclusion criteria, women were invited to participate in the trial and to sign an informed consent form.

After acceptance to participate and prior to randomization, all the participants were asked about their beliefs regarding the effectiveness of honey to ensure that the women’s beliefs about the effect of honey would not be influenced by their knowledge of their randomization group since it was not possible to blind them from it. We asked the following question: “Do you think that applying honey to wounds or scars is helpful?”. To assess the participants’ responses, we used a five-point Likert scale (not at all, a little, I do not know, a lot, or very much).

### 2.4. Interventions

The women included in the intervention group received Medihoney^®^ (Brisbane, QLD, Australia) wound gel, which is made of 80% Manuka honey and 20% wax. Medihoney^®^ is authorized on the pharmaceutical market in Switzerland and is part of the wound care protocol at the University Hospitals of Geneva [23]. This honey is known for its antibacterial activity, especially against many pathogens such as Staphylococcus aureus and Escherichia coli [24]. Given the little literature about the use of honey in the perineal area, we referred to the protocol of a Swiss maternity hospital that commonly uses honey in its practice. We also considered recommendations developed for other wound types [25,26]. Recommendations are to store Medihoney^®^ wound gel at room temperature, with a maximum shelf life of one week after opening [27]. Participants in the intervention group were asked to apply honey to the entire perineal trauma area after usual toileting practices, with no vaginal intrusion, as often as possible, but at least twice a day during the first five days postpartum (from D0 to D4). In addition, they received standard maternity care (i.e., hygiene advice with hand washing and perineal cleaning, standard pain management such as paracetamol and ibuprofen, use of cold packs, body positioning for perineal discharge, and diet advice). They were allowed to perform as much perineal hygiene as they wished between applications of honey.

The women in the control group received the same standard care as the intervention group.

### 2.5. Outcomes

Their beliefs regarding the effectiveness of honey were compared between groups. Sociodemographic and obstetrical data were collected. The primary outcome was perineal pain intensity evaluated using the Visual Analog Scale (VAS, 0 = no pain to 10 = the worst possible pain) on Day 1 (D1) and Day 4 (D4) postpartum. The qualitative aspect of the pain was assessed using the Questionnaire de Douleur Saint-Antoine (QDSA), which is a short French version of the McGill Pain Questionnaire [28,29]. This questionnaire consists of a simple verbal scale that describes the intensity of the pain (no pain, mild, discomforting, distressing, horrible, and excruciating). It also includes a list of 15 adjectives that describe the sensory and emotional aspects of pain. Five response modalities (none, mild, moderate, strong, and severe) are used to assess the perceived intensity of each adjective. It is possible to calculate a pain intensity score with this questionnaire.

The secondary outcomes were the burning sensation on wounds during urination because contact with urine may increase burning sensation and pain, particularly in the presence of grazes. We assessed it with the VAS (0 = no burn sensation to 10 = the worse burning sensation) on D1 and D4 postpartum. The number and dosage of pain relief drugs taken over 24 h were also recorded in order to calculate a daily dose.

Women in both groups were asked to complete the same questionnaire in the evening on D1 and D4 postpartum. Questionnaires were returned in prepaid envelopes. Reminders were sent via telephone, emails, or messages one month after inclusion in the case of non-return to limit the responses lost to follow-up. In the case of no response, we considered them as lost to follow-up.

For the intervention group only, we assessed participants’ satisfaction with the application of honey using a five-point Likert scale (not at all, a little satisfied, somewhat satisfied, satisfied, and very satisfied). We also recorded the number of daily applications of honey to assess compliance with the protocol of the intervention group.

### 2.6. Statistical Analysis

Data analysis and reporting were performed according to the CONSORT guidelines for randomized controlled trials. The analysis was conducted according to the intention-to-treat principle. Analyses were performed using the STATA software package version 15 (StataCorp, College Station, TX, USA, 2017). The proportions between the groups were compared using the Chi-Square test or Fisher’s exact test if appropriate. The continuous data were compared using Student’s t-test or the Mann–Whitney U test if the distribution was skewed. Kruskal–Wallis tests were used to compare the means between several groups. The associations were considered statistically significant when the *p*-value was below 0.05 (probability of type I error = 0.05). We evaluated the intensity score of the QDSA based on the sum of the intensity scores provided by the women. Data were stored in a relational database management system using the dedicated clinical database management system software REDCap^TM^, a mature, secure web application for building and managing online surveys and databases.

## 3. Results

### 3.1. Sociodemographic and Clinical Characteristics of Participants

Among 99 women invited to participate, 68 provided written informed consent and were randomized, and 60 (83.8%) returned the questionnaires (Figure 1). Three women in the control group and five in the intervention group never returned their questionnaires. When contacted, they said that this was due to the loss of the questionnaire, a lack of time, breastfeeding problems, or difficulties related to the use of honey without precision. The sociodemographic and clinical characteristics of participants are reported in Table 1. There was no significant difference between the two groups regarding the women’s beliefs about the effectiveness of honey on wounds prior to randomization (Table 2).

### 3.2. Comparison of the Primary and Secondary Outcomes between Groups

There was no significant difference (Table 3) in the measure of the intensity of pain using the VAS between the control group and the intervention group on Day 1 (VAS 3.38 vs. 3.34, respectively, *p* = 0.65) or Day 4 (VAS 2.28 vs. 1.41, respectively; *p* = 0.09). Secondary analysis on stratification for grazes was not statistically different on D1 or D4, although there was a trend on D4 in favor of using honey with the presence of grazes (*p* = 0.05). Responses from the QDSA Likert scale showed no significant difference in pain sensation between the two groups on D1 (*p* = 0.95) and D4 (*p* = 0.44) (Figure 2).

There was no significant difference in the average experience of urinary burning sensations between groups on D1 and D4. A secondary analysis of stratification for grazes showed no statistically significant differences either on D1 or D4 when comparing both groups. However, on D4, women with grazes in the control group experienced more urinary burning than women without grazes in the same group (VAS 2.69 vs. 1.14, respectively, *p* = 0.03).

The number of women using pain relief drugs was not significantly different between groups on D1 and D4. We observed a significant decrease in the consumption of Ibuprofen^®^ between groups on D4 in favor of the intervention group (*p* = <0.05).

### 3.3. Compliance and Satisfaction with the Intervention

Women in the intervention group were either satisfied or very satisfied (93%) with the honey application on their perineal lesions on Day 1 and Day 4 (Table 4). Two women were little or somewhat satisfied with the application of honey on D1 and D4. Dissatisfaction was related to the difficulty in applying honey to edematous perineum, an unpleasant sensation due to the sticky texture of the honey, and difficulties in touching the perineum after birth.

Regarding compliance on D1, most women applied honey twice a day or more (28/29, 96.6%), with a range of one to five applications and a mean of 2.7 applications per day. One woman did not apply honey at least twice on D1 because she did not understand the protocol and applied the entire honey tube in one application. Regarding D4, 28/29 of the women (96.6%) applied honey at least twice a day, with a range of two to four applications and a mean of 2.6 times per day. One woman did not answer this question on D4 and provided no explanation as to why.

## 4. Discussion

This RCT was carried out to evaluate the effectiveness of the application of honey in reducing pain resulting from perineal damage during the early postpartum period. We could not demonstrate a significant decrease in perineal pain between groups on D1 and D4, even though the mean VAS was lower in the intervention group on D4. Lavaf et al. [13], the only trial that studied the same topic, also failed to show a decrease in perineal pain on D7 and D10. We were surprised by the low level of pain reported by the women. It was much lower than expected based on other studies on perineal pain [15,16].

Even if the use of honey did not significantly reduce the burning sensation during urination between groups, women in the control group experienced a significantly worse urinary burning sensation in the presence of grazes. This result supports the hypothesis that the presence of grazes may increase burning sensations during urination and therefore justified the stratification based on this variable. It should be noted that Lavaf et al. did not investigate the impact of grazes in their study [13].

Participants who applied honey consumed significantly less ibuprofen than others on D4. Lavaf et al. did not find this difference [13]. Shirvani et al. noted a decrease in the intake of pain relief drugs during the first 10 days post-cesarean section (11.5% for the honey group compared to 62.6% for the placebo group and 45.9% for the control group, *p* = 0.02) [30]. We must interpret these results cautiously because the use of pain relief drugs during the postpartum period is proposed to treat pain of various origins, and our sample size was not calculated with this variable in mind. Further analysis is therefore required [9].

Most women were either satisfied or very satisfied with the use of honey during both measurement periods. In addition, compliance with the protocol remained high and stable between D1 and D4. This demonstrates the acceptability of applying honey over several days. Before our study, satisfaction with the use of honey on the perineum had not been evaluated in scientific studies. No participant reported severe side effects from the use of honey in our study. These results are identical to those of a previous literature review [31].

Some methodological elements, such as the timing of the first honey application, the application time, the duration of the application, the timing of the pain assessment, and the type of honey used, may have impacted our results. The first application of honey ranged from 2 to 24 h postpartum and was comparable with the study of Lavaf et al. [13]. Regarding the ideal time for the application, the participants in Lavaf et al.’s study [13] used honey in the evening from bedtime until D10. We chose to ask women to use honey at least twice a day until D4, in accordance with other studies [30,32]. Honey is a viscous, thick product that protects the wound and supports the healing process, but it can flow more easily from the perineal area because it is not possible to apply a dressing. Therefore, if honey is applied as early as possible and several times a day, it might have a better effect [33]. The evaluation of pain on D1 was questionable, as the delay for intervention (honey) was too short to show any benefit. According to a systematic review, the incidence of perineal pain two days after vaginal birth differs relatively little between women with intact perineum (42%), first-degree perineal tears (64.3%), and second-degree perineal tears (39.0%) [34]. However, there is a greater difference between D4 and D10 (5%, 25.5%, and 23%, respectively). Thus, in future studies, we recommend evaluating pain beginning on D4.

The variety of honey used should also be considered, as the properties may vary from one type to another [24]. Lavaf et al. used sterilized natural honey from the Qamsar region of Iran [13]. However, Heidari et al. used honey from the Iranian plant Astragalus gossypinus for cesarean section scars [32]. In our study, we used Medihoney^®^, a Manuka honey-based product known for its medicinal properties [24,35].

There were several strengths and limitations in this study. We conducted an RCT, which is the best method for evaluating the effectiveness of an intervention. We obtained a very good response rate for our pain evaluation questionnaires. Moreover, the subject is innovative because it is the first European study to evaluate both the effectiveness of honey in terms of pain relief in perineal lesions and women’s satisfaction with its use. The use of a honey-like placebo would have been a strength, but we were confronted with financial and time constraints, and such a placebo would be difficult to create because of honey’s particularities. The responses concerning the participants’ belief in the effects of honey on wounds showed that we could control the women’s representations. If these beliefs had been different between groups, it could have biased the responses to the measured variables. It is necessary to highlight that pain is a subjective experience with multidimensional aspects (affective–emotional, sensory, cognitive, and behavioral) [36]. The perinatal period is a time of emotional intensity and psychological vulnerability, as various psychological changes take place [37]. In addition to measuring the participants’ perception of pain, it would have been interesting to repeat this study by evaluating the effect of honey on the healing process using a grid such as that of REEDA [18]. This variable would be more objective than perceived pain. The type of wound suture used could impact the feeling of pain [38], but all women included in this study had overjet suturing, according to the institutional protocol. We only stratified our subjects based on the presence of grazes. In future studies, it would be interesting to calculate sufficient statistical power to measure the effect of this. Finally, our sample size was too small to show any benefits of the use of honey for reducing perineal pain, as the VAS measurement showed lower values than expected. According to our study’s VAS results, a sample size of 83 women in each group would be necessary to obtain a significant difference for this primary outcome [19].

## 5. Conclusions

This randomized controlled trial was unable to show a benefit from the application of honey on perineal lacerations to reduce the intensity and perception of perineal pain in the early postpartum period. Despite this, most of the women included in the intervention group were satisfied or very satisfied with the use of honey, and we observed a lower consumption of painkillers on D4 in the group using honey.

## Figures and Tables

**Figure 1 healthcare-10-01515-f001:**
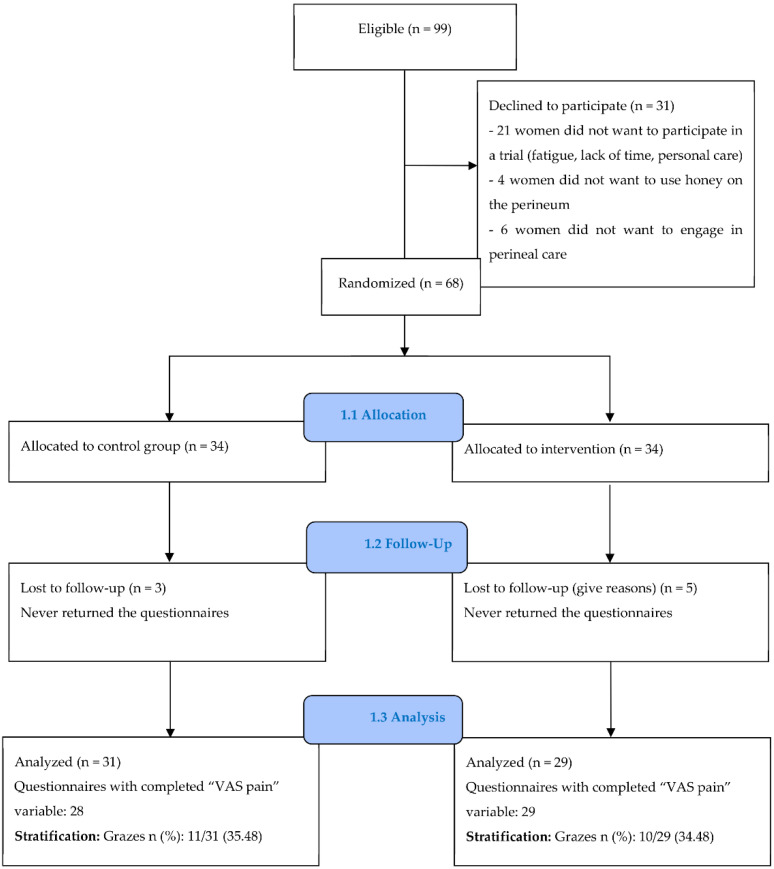
CONSORT flow chart.

**Figure 2 healthcare-10-01515-f002:**
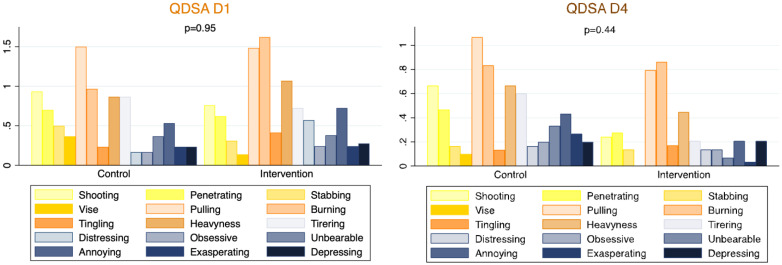
Questionnaire De Saint Antoine (QDSA) responses.

**Table 1 healthcare-10-01515-t001:** Baseline sociodemographic and clinical characteristics.

		Control (*n* = 31)	Intervention (*n* = 29)	Total (*n* = 60)
Mean age (SD)		33.80 (5.04)	33.31 (4.31)	33.57 (4.67)
Mean BMI (SD)		23.54 (3.87)	24.07 (3.74)	23.80 (3.79)
Education *n* (%)	Primary school	2/31 (6.45)	3/29 (10.34)	5/60 (8.33)
	Secondary school	4/31 (12.90)	2/29 (6.90)	6/60 (10.00)
	Apprenticeship	7/31 (22.58)	4/29 (13.79)	11/60 (18.33)
	University	18/31 (58.67)	20/29 (68.97)	38/60 (63.33)
Tobacco use *n* (%)		4/31 (12.90)	3/29 (10.34)	7/60 (11.67)
Diabetes ^1^ *n* (%)		0/31 (0.00)	4/29 (13.79)	4/60 (6.67)
Gestation *n* (%)	1st	13/31 (41.94)	12/29 (41.38)	25/60 (41.67)
	≥2	18/31 (58.06)	17/29 (58.62)	35/60 (58.33)
Parity *n* (%)	1st	17/31 (54.84)	14/29 (48.28)	31/60 (51.67)
	≥2	14/31 (45.16)	15/29 (51.72)	29/60 (48.33)
Personal treatment ^2^ *n* (%)		4/31 (87.10)	4/29 (86.21)	8/60 (13.33)
Mode of birth *n* (%)	Spontaneous ^3^	23/31 (74.19)	22/29 (75.86)	45/60 (75.00)
	Instrumental ^4^	8/31 (25.81)	7/29 (24.14)	15/60 (25.00)
Anesthesia *n* (%)	Epidural	25/31 (80.65)	19/29 (65.52)	44/60 (73.33)
	Local	6/31 (19.35)	10/29 (34.48)	16/60 (26.67)
Perineal tears *n* (%)	1st degree	12/31 (38.71)	13/29 (44.83	25/60 (41.67)
	2nd degree	15/31 (48.39)	10/29 (34.48)	25/60 (41.67)
	Episiotomy	4/31 (12.90)	6/29 (20.69)	10 /60 (16.67)
Grazes *n* (%)		11/31 (35.48)	10/29 (34.48)	21/60 (35.00)
Mean neonate weight gr (SD)		3285 (570)	3357 (350)	3320 (474)

^1^ Limited to gestational diabetes. ^2^ Anti-asthmatics, thyroid hormones, and anti-epileptics. ^3^ Vaginal childbirth without vacuum or forceps. ^4^ Vaginal childbirth using vacuum or forceps.

**Table 2 healthcare-10-01515-t002:** Representation of the effectiveness of honey before randomization.

	Control (*n*= 31)	Intervention (*n* = 29)	Chi 2
Not at all *n* (%)	0 (0.00)	0 (0.00)	
A little *n* (%)	6 (19.35)	6 (20.69)	
I do not know *n* (%)	8 (25.81)	13 (44.83)	0.33
A lot *n* (%)	14 (45.16)	7 (24.14)	
Very much *n* (%)	3 (9.68)	3 (10.34)	

**Table 3 healthcare-10-01515-t003:** Comparison primary and secondary outcomes between two groups at Day 1 and 4 postpartum.

			Day 1			Day 4		
			Control	Intervention	*p*-Value	Control	Intervention	*p*-Value
VAS perineal pain		Mean VAS pain (SD)	3.38 (2.14) (*n* = 29)	3.34 (2.35) (*n* = 29)	0.65	2.28 (1.96) (*n* = 28)	1.41 (1.49) (*n* = 29)	0.09
		Mean without grazes (SD)	3.61 (2.36) (*n* = 18)	3.76 (2.48) (*n* = 19)	0.87	1.84 (1.67) (*n* = 18)	1.5 1.77) (*n* = 19)	0.57
		Mean with grazes (SD)	3.02 (1.78) (*n* = 11)	2.53 (1.96) (*n* = 10)	0.50	3.08 (2.26) (*n* = 10)	1.15 (0.78) (*n* = 10)	0.054
		*p*-value	0.49	0.22	-	0.49	0.22	-
QDSA	Likert QDSA	No pain *n* (%)	2/31 (6.45)	1/29 (3.45)	0.95	5/30 (16.67)	6/29 (20.69)	0.44
		Mild *n* (%)	9/31 (29.03)	8/29 (27.59)		12/30 (40.00)	15/29 (51.72)	
		Discomforting *n* (%)	15/31 (48.39)	15/29 (51.72)		12/30 (40.00)	7/29 (24.14)	
		Distressing *n* (%)	5/31 (16.13)	5/29 (17.24)		0/30 (0.00)	1/29 (3.45)	
		Horrible *n* (%)	0/31 (0.00)	0/29 (0.00)		1/30 (3.33)	0/29 (0.00)	
		Excruciating *n* (%)	0/31 (0.00)	0/29 (0.00)		0/30 (0.00)	0/29 (0.00)	
	Sensory category	Mean intensity score (SD)	6.07 (4.55)	6.41 (4.19)	0.64	4.10 (4.40)	2.93 (2.83)	0.44
		Mean number of words chosen (SD)	3.53 (2.24)	3.79 (2.01)	0.63	2.6 (2.08)	2.17 (1.65)	0.47
	Affective category	Mean intensity score (SD)	2.57 (3.56)	3.14 (3.54)	0.30	2.2 (3.86)	1.0 (1.89)	0.20
		Mean number of words chosen (SD)	1.53 (2.03)	2.28 (2.28)	0.14	1.47 (2.24)	0.86 (1.51)	0.24
	Total	Mean intensity score (SD)	8.63 (7.64)	9.55 (7.03)	0.42	6.3 (7.94)	3.93 (3.90)	0.46
		Mean number of words chosen (SD)	5.07 (3.92)	6.07 (3.96)	0.33	4.07 (4.01)	3.03 (2.85)	0.34
Pain relief drugs use		Women who used pain relief drugs *n* (%)	26/31 (83.87)	23/29 (79.31)	0.75	17/31 (54.84)	13/28 (46.43)	0.61
		Mean paracetamol use in mg (SD)	1466.67 (1224.28)	1535.71 (1393.96)	0.94	866.67 (1129.03)	611.11 (923.34)	0.50
		Mean ibuprofen use in mg (SD)	646.67 (537.38)	707.14 (614.59)	0.79	466.67 (539.05)	192.59 (339.60)	0.049
VAS urinary burning		Mean VAS urinary burning (SD)	2.42 (2.18) (*n* = 29)	2.64 (2.52) (*n* = 28)	0.99	1.73 (2.07) (*n* = 29)	1.32 (1.93) (*n* = 28)	0.31
		Mean without grazes (SD)	2.23 (2.41) (*n* = 18)	2.50 (2.69) (*n* = 18)	0.94	1.14 (1.52) (*n* = 18)	1.39 (2.22) (*n* = 18)	0.94
		Mean with grazes (SD)	2.73 (1.80) (*n* = 11)	2.88 (2.29) (*n* = 10)	0.97	2.69 (2.55) (*n* = 11)	1.19 (1.34) (*n* = 10)	0.12
		*p*-value	0.29	0.45	-	0.03	0.71	-

**Table 4 healthcare-10-01515-t004:** Degree of satisfaction of the application of honey (intervention group).

	Day 1		Day 4	
Degree of Satisfaction	*n* = 29		*n* = 28	
Not at all *n* (%)	0/29 (0.00)	2/29 (6.90)	0/28 (0.00)	2/28 (7.14)
A little satisfied *n* (%)	0/29 (0.00)		0/28 (0.00)	
Somewhat satisfied *n* (%)	2/29 (6.90)		2/28 (7.14)	
Satisfied *n* (%)	18/29 (62.07)	27/29 (93.10)	11/28 (39.29)	26/28 (92.86)
Very satisfied *n* (%)	9/29 (31.03)		15/28 (53.57)	

## Data Availability

Not applicable.

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
