# Peer review of "Application of Honey to Reduce Perineal Laceration Pain during the Postpartum Period: A Randomized Controlled Trial"

_healthcare, 2022, doi:10.3390/healthcare10081515_

Round 1

Reviewer 1 Report

The discussed problem of postpartum perineal pain is a problem which affects many women but is underestimated by medical providers. The construction of the study is appropriate and consists of all necessary parts. Nevertheless, some issues need further explanation.

The introduction section is a well-constructed part of the study. The problem of perineal lesions is appropriately introduced to the reader. But there is luck information about the honey specifications. I would recommend adding more information and citations in this field.

The aim of the study is well established and presented. I also have no reservations about the Methodological part and used statistical methods. The study is conducted according to PRISMA guidelines for observational studies. PICO question is well described, and the inclusion and exclusion criteria are well presented. Nevertheless, I recommend adding the PRISMA flow chart as a supplementary file.

It is exciting how honey, as a highly sugar-including product, affects the occurrence of infections in the wound after perianal lesions. Did the authors consider it?

In lines 246-249, you said, “…Ibuprofen passes into breast milk somewhat and, like any medication, is associated with possible side effects…” I am unsure if this thought is appropriate. According to many studies, no harmful influence of ibuprofen was found, and as you admit, honey usage was not correlated with pain relief. Publication of such information could be detrimental to breastfeeding women, who are afraid to harm their children. Perineal pain with anxiety and not taking painkillers could worsen maternal-child relations. Therefore, I would recommend deleting this sentence or paraphrasing it.

Reviewer 2 Report

Dear authors, thank you for carrying out the research, here are some suggestions for improvement in your manuscript:

- In line 28 you refer to the prevalence of injuries, but you do not make it clear where, is it in Switzerland, in the world, Europe???? you should clarify.

- In line 64 you make reference to the fact that the trial was carried out between 2018 and 2019, why have you not developed the manuscript until now? in these years a new drug, antibacterial or better product than the one you are proposing could have come out?

- From line 68 to 72 does not refer to the paragraph in which it is written, check.

- You should improve the material and methods section ostensibly. You should indicate where the study was carried out. In addition, they should indicate a clinical trial registration number, either in US or European clinicaltrial. They should also indicate that the intervention was conducted according to the guidelines of the Declaration of Helsinki.

- Lines 308 and 309 should be deleted.

- The conclusion of the manuscript is very generic and without contributions.

- In my view, they start from a biased theory in the research and take into account an aspect that they should demonstrate and refer to in line 68-70. Furthermore, the statistical tables do not show any significance in the intervention between the groups, therefore it is a study lacking scientific evidence and is only based on the subjective feeling of the patients, which may be biased in this intervention. 

I invite you to thoroughly improve your manuscript. Regards.

Reviewer 3 Report

The authors performed a randomized, controlled study investigated the use of honey in the treatment of perineal pain postpartum. This is a well-conducted study with an interesting concept that has good potential for investigating honey as topical analgesic in postpartum pain management.  Please find attached my comments below:

1. In the introduction section, please elaborate on the risk factors for developing perineal and vaginal tears. Suggested reference (PMID: 33267813)

2. Page 2, line 50-52; please provide an appropriate reference.

3. Page 2, line 59-60; please define "Western context"

4. Page 3, section 2.5 Interventions: please specify whether women in the intervention group were allowed to perform any perineal hygiene in between honey application.

5. You also mention spontaneous and instrumental vaginal deliveries. Can you specify what type of instrumental mode of birth were used? In terms of episiotomies, please specify the ratio between midline or lateral incision points.

Reviewer 4 Report

The manuscript present a very important topic in obstetrics.

However, the text and the study itself must be improved before publication.

The title is not clear, please reconsider - suggested "Application of honey to Reduce Perineal Laceration Pain during the Postpartum Period: A Randomized Controlled Trial" or any other by authors, but more clear.

The abstract requires English editing to become more clear. Example of suggested rewriting: "Rates of Perineal lesions following vaginal birth range from 53 to 79%. They may be responsible for adverse outcomes for women. Honey has demonstrated interesting properties to promote wound healing. The aim of this study was to evaluate the effectiveness of honey in reducing l perineal pain related to lacerations after childbirth." The editing should be done through the whole abstract text. Moreover, the abstract must be improved in general to properly represent the study. 

The introduction part is too narrow and does not provide a full rationale of the study. Please include your local statistics on birth rates and lacerations after childbirth, including complications after lacerations and adverse outcomes rates (since you've mentioned in the abstract that perineal lacerations are responsible or adverse effects).

Please state your aim clearly at the end of the introduction part. 

In the methods part, please justify your sample size and provide the calculations that suggest with this small sample size you could get a statistically significant data. 

The results are supported by clear tables and figures. However, it is a doubtful, if with the such small sample sizes per groups, statistically significant result could be obtained.

The conclusion part should be rewritten to provide a clear and justifies findings of the study, written in an academic style appropriate for a research manuscript.  Lines 308 and 309 of the conclusion are confusing.

A thorough English editing should be done through the text, as in the current version some sentences are confusing and difficult to comprehend. 

Round 2

Reviewer 2 Report

Dear authors:

Thank you for making a substantial change to your manuscript, it is much improved, and although there are possible improvements, there are no substantial flaws. Thank you for adding the NCT number.

Best regards.

Author Response

Thank-you for your appreciation.

We added the NCT number in "Institutional Review Board Statement"  as  www.kofam.ch / n°SNCTP000002950 (highlighted in blue)

Reviewer 4 Report

Thank you very much or the changes performed

Author Response

Thank you very much for your appreciation.